# CluB: Cluster Meets BEV for LiDAR-Based 3D Object Detection

**Yingjie Wang**[1,2*]**, Jiajun Deng**[3†]**, Yuenan Hou**[2]**, Yao Li**[1]**, Yu Zhang**[1]**,**

**Jianmin Ji**[1]**, Wanli Ouyang**[2]**, Yanyong Zhang**[1†]

[1]University of Science and Technology of China,

[2]Shanghai AI Laboratory, [3]The University of Adelaide, AIML

## Abstract

Currently, LiDAR-based 3D detectors are broadly categorized into two groups, namely, BEV-based detectors and cluster-based detectors. BEV-based detectors capture the contextual information from the Bird's Eye View (BEV) and fill their center voxels via feature diffusion with a stack of convolution layers, which, however, weakens the capability of presenting an object with the center point. On the other hand, cluster-based detectors exploit the voting mechanism and aggregate the foreground points into object-centric clusters for further prediction. In this paper, we explore how to effectively combine these two complementary representations into a unified framework. Specifically, we propose a new 3D object detection framework, referred to as CluB, which incorporates an auxiliary cluster-based branch into the BEV-based detector by enriching the object representation at both feature and query levels. Technically, CluB is comprised of two steps. First, we construct a cluster feature diffusion module to establish the association between cluster features and BEV features in a subtle and adaptive fashion. Based on that, an imitation loss is introduced to distill object-centric knowledge from the cluster features to the BEV features. Second, we design a cluster query generation module to leverage the voting centers directly from the cluster branch, thus enriching the diversity of object queries. Meanwhile, a direction loss is employed to encourage a more accurate voting center for each cluster. Extensive experiments are conducted on Waymo and nuScenes datasets, and our CluB achieves state-of-the-art performance on both benchmarks.

## 1 Introduction

3D perception on point clouds has attracted much attention from both industry and academia thanks to its wide applications in various fields such as autonomous driving and robotics [29, 17, 11, 28, 32]. LiDAR-based 3D object detection is one of the vital tasks, which takes sparse point clouds as input to estimate the 3D positions and categories of objects [20, 7, 30, 40].

Currently, the mainstream 3D object detectors [35, 21, 18, 38, 22, 1, 39] convert unstructured point clouds into regular grids in the Bird's Eye View (BEV) for feature representation (Figure 1 (a)), named as BEV-based 3D object detectors. The regular grids in BEV are beneficial to feature extraction and contextual information capturing. As the key point to make a prediction, *i.e.*, the center point of an object, can be empty, the features of the center points are often diffused from the surrounding points via a stack of convolution layers. This practice significantly weakens the capability of presenting an object with the center point.

---

∗ This work was done during Yingjie's internship at Shanghai Artificial Intelligence Laboratory.

† Corresponding Authors: Jiajun Deng and Yanyong Zhang.

37th Conference on Neural Information Processing Systems (NeurIPS 2023).

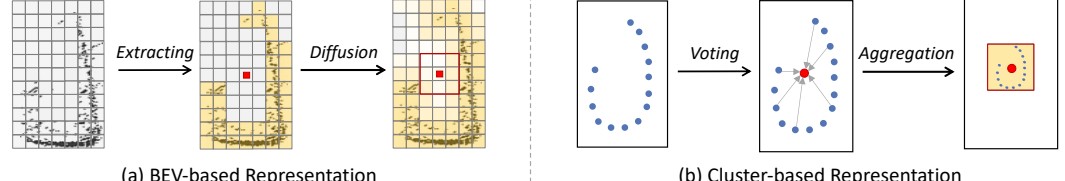

(a) BEV-based Representation          (b) Cluster-based Representation

Figure 1: Illustration of (a) *context-aware* BEV-based representation[10] and (b) *object-centric* cluster-based representation. The BEV representation can capture the rich contextual information of an object. However, the missing feature of the center point (red square), which is also the key to making the prediction, is diffused from occupied voxels. The cluster-based representation is generated using the voting mechanism by aggregating the foreground points into the object-specific feature for the voting center (red dot). This representation mainly focuses on the foreground points and fails to utilize the contextual information.

Another stream of detectors [10, 19, 13, 36] resorts to the cluster-based feature representation (Figure 1 (b)). This line of methods exploits the voting mechanism and groups the foreground points into object-centric clusters for further prediction. The cluster features largely preserve the 3D structure details of each object, while avoiding detection based on center points that are absent in the raw data. However, the clustering operation mainly focuses on foreground points, and fails to fully utilize the contextual information. It heavily hinges on foreground segmentation accuracy. Based on the potential synergy between these two paradigms, in this paper, we focus on exploring the combination of context-aware BEV representation and object-centric cluster representation to improve the accuracy of 3D object detection, which remains under-explored in the literature.

Towards this goal, an intuitive solution is to scatter all cluster features to the BEV space based on the locations of the predicted voting centers[1] and directly concatenate the obtained vote BEV features (vote BEV in short) to the dense BEV features (dense BEV in short). However, the established hard association[2] between cluster features and BEV features heavily relies on the quality of vote clusters, not to mention that the representation of these two streams is not well aligned, which may affect the overall stability of representation learning [5]. This potential issue makes the intuitive strategy fail to fully explore the feature intertwining of both representations.

Therefore, by consolidating the idea of leveraging the object-centric property of cluster representation and contextual information of BEV representation, we propose a novel 3D object detection framework, *i.e.*, **CluB**, which integrates an auxiliary **Clu**ster-based branch to the mainstream **B**EV-based detector by enriching the object representation at both feature and query levels. Specifically, CluB begins with projecting the two features into the BEV space (vote BEV and dense BEV) for unifying the object representation from corresponding branches. Then, we propose a two-step approach to enhance the representation capability from the two feature and query aspects.

Firstly, to address the issue of the hard association between cluster features and BEV features, we propose a Cluster Feature Diffusion (CFD) module that adaptively diffuses the valid votes on the vote BEV to their neighboring regions according to the predicted class of each vote, which repositions the association from hard to soft and adaptive. Based on that, an imitation loss is introduced to transfer object-centric knowledge to the BEV branch and encourage the stability of overall representation learning. In this way, the object representation can be jointly enhanced by fusing features from both branches.

Secondly, since most of the BEV detectors detect objects as center points, it is non-trivial to enrich the diversity of object queries using the voting centers directly from the cluster branch. We thus design a Cluster Query Generation (CQG) module, which activates and then selects the top-ranked candidates based on the vote BEV. The top-selected cluster queries are then appended to the original object queries together for better prediction. On the one hand, we also utilize direction supervision to alleviate the overlap problem for close clusters, thus producing more accurate cluster queries.

---

[1]The expression of the vote center is referred to as VoteNet [19]. VoteNet processes existing points to generate votes, which serve as the cluster centers.

[2]In fact, the vote centers indicate the potential locations of the centers of objects within a certain area.

The design of CluB strengthens the object representation through feature integration and query expansion, eventually bringing significant performance gains. In summary, we make the following contributions:

- We develop the CluB framework to improve the accuracy of 3D object detection by taking advantage of both BEV-based and cluster-based paradigms.

- We present an elegant solution to integrating the context-aware BEV representation and object-centric cluster representation at both the feature level and the query level, which is a non-trivial problem.

- Our approach outperforms state-of-the-art methods by a remarkable margin on two prevalent datasets, *i.e.*, Waymo Open Dataset and nuScenes, demonstrating the effectiveness and generality of our proposed method.

## 2 Related Work

**BEV-based LiDAR Detectors.** There are mainly two ways to convert the point cloud into BEV representation. One is to extract BEV features in 2D space and design efficient networks to produce predictions based on the BEV features [15, 21, 9]. PointPillars [15] introduces the pillar representation (a particular form of the voxel) and process it with 2D convolutions. PillarNet [21] resorts to the powerful encoder network for feature extraction and designs the neck network to achieve spatial-semantic feature fusion, thus greatly improving the detection performance of pillar-based models. The other way is to extract point cloud features in 3D space and scatter them to the BEV plane, yielding more accurate detection results [37, 33, 35]. With the BEV representation, CenterPoint [35] designs an anchor-free one-stage detector, which extracts BEV features from voxelized point clouds to estimate object centers and all object properties such as 3D size and orientation. Later, the adoption of the popular transformer architecture as the detection head has substantially improved the accuracy and robustness of BEV-based methods [38, 1, 27, 39]. SWFormer [27] applies the window-based transformer on LiDAR BEV features. In particular, SWFormer solves the center feature missing problem for the 3D sparse features on the BEV planes with the voxel diffusion strategy, while not considering the semantic consistency. Transfusion [1] designs an effective and robust transformer-based LiDAR-camera fusion framework. which can also serve as a strong LiDAR-based 3D detector featured by the BEV-based paradigm.

**Cluster-based LiDAR Detectors.** The cluster-based methods focus on extracting discriminative features directly from clustered point clouds and have shown promising results in various benchmarks [10, 19, 13, 36]. VoteNet [19] is a pioneering work in the field of cluster-based LiDAR detectors. It employs the voting mechanism to predict object centers from the input point cloud. Motivated by that, FSD [10] designs the fully sparse network in the outdoor scene using point clustering and group correction. PSA-Det3D [13] proposes a pillar set abstraction method (PSA) to learn representative features for sparse point clouds of small objects, which mainly benefits from point-wise feature aggregation. Current benchmarks [26, 2, 31] are dominated by the aforementioned BEV-based and cluster-based 3D detectors. Notably, a coherent work VoxelNext [4] predicts objects directly based on sparse voxel features, without relying on cluster proxies or the dense BEV representation.

In summary, cluster-based detectors are specialized at capturing fine-grained features for individual objects consisting of foreground points, while BEV-based detectors are well-suited for capturing contextual information of point clouds but may have weaker center point representation capabilities. Therefore, by combining these approaches, we can leverage their respective advantages to enhance the detection performance. Our method aims to integrate a BEV-based detector with an auxiliary cluster-based branch, thereby harnessing the strengths of both paradigms to enhance the overall detection performance.

## 3 Methodology

In this section, we introduce the proposed CluB in detail, as shown in the Figure 2.

**Overview.** Our CluB is built from scratch using the following three steps. First, we utilize a sparse voxel encoder to extract voxel features and aggregate the vote cluster features, which is considered

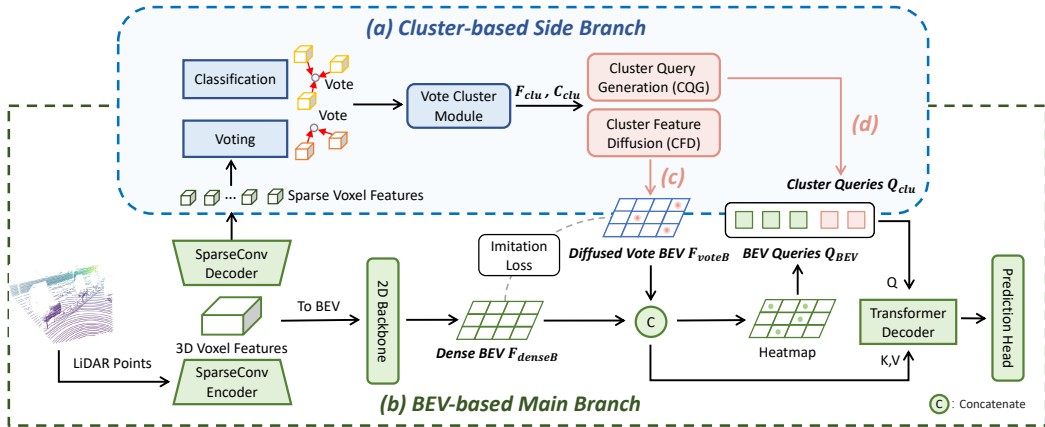

Figure 2: The overall framework of CluB. The cluster-based side branch (a) is integrated into the BEV-based main branch (b) through a two-step scheme, *i.e.*, feature-level integration (c) and query-level expansion (d). Specifically, the raw point clouds are first voxelized and then fed into the sparse U-Net to extract both 3D voxel features and sparse voxel features. The former is used to obtain the dense BEV feature. The latter is processed in the cluster branch to generate a set of vote cluster features $\mathbf{F}_{clu}$ and corresponding voting centers $\mathbf{C}_{clu}$. Based on the two outputs, *i.e.*, $\mathbf{F}_{clu}$ and $\mathbf{C}_{clu}$, the Cluster Feature Diffusion (CFD) module obtains a diffused vote BEV feature $\mathbf{F}_{voteB}$ to enhance the dense BEV feature $\mathbf{F}_{denseB}$ supervised by an imitation loss, and the Cluster Query Generation (CQG) module outputs a set of top-ranking cluster queries $\mathbf{Q}_{clu}$, which are concatenated to the BEV queries $\mathbf{Q}_{BEV}$ obtained from the heatmap. The decoder takes hybrid object queries as input and makes the final predictions based on the concatenated BEV features.

as the cluster-based auxiliary branch (Section 3.2). Next, a BEV-based main branch (Section 3.1) is served as the detection backbone. Finally, the cluster-based side branch is incorporated into the BEV-based main branch through a two-level enhancement scheme, *i.e.*, feature-level integration (Section 3.3) and query-level expansion (Section 3.4).

## 3.1 BEV-based Main Branch

The main BEV branch takes the voxelized 3D point clouds as input and then leverages a sparse encoder consisting of sparse convolution layers to produce 3D voxel features [24]. Then we compress the 3D voxel features into 2D feature maps in the BEV space. The 2D BEV feature map is fed to the 2D convolution-based backbone to generate a dense BEV feature map $\mathbf{F}_{denseB} \in \mathbb{R}^{C \times H \times W}$ (short for dense BEV), where $(H, W)$ indicate the height and width of the feature map. Compared to the anchor/center-based detection head [35, 37], the transformer decoder demonstrates great potential for feature fusion besides the notable performance improvement [34, 8]. We thus utilize a transformer-based detection head [1] to extract class-level feature representations from dense BEV for object prediction. The decoder layer follows the design of DETR [3]. We generate object candidates from the class-specific heatmap and select the top-ranked candidates for all the categories as our initial BEV queries $\mathbf{Q}_{BEV}$. The fused BEV features are also used as key-value sequences for the transformer decoder.

## 3.2 Cluster-based Auxiliary Branch

The cluster side branch takes the same voxelized 3D point clouds as input to extract sparse voxel features via a sparse convolution based encoder-decoder. These sparse voxel features are fed into two heads for foreground classification and object center voting. Based on the voting results, we group the foreground points into clusters using the grouping strategy (*e.g.*, ball query), and then use a vote cluster module to aggregate each cluster feature. The vote cluster module could be implemented with simple vote aggregation following [19]. In order to extract more powerful vote cluster features, we utilize a highly efficient sparse instance recognition module for further extraction [10]. The final

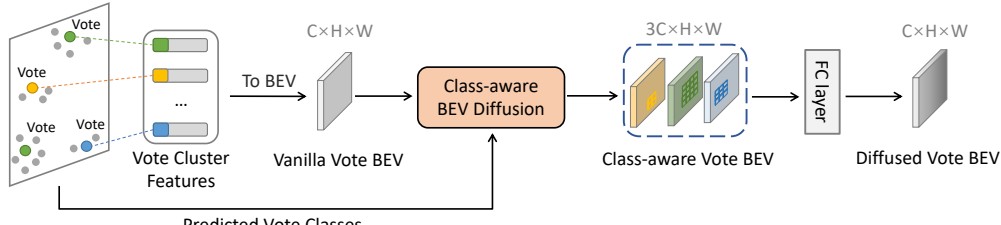

Figure 3: Illustration of Cluster Feature Diffusion (CFD) module. We first transform the vote cluster features to the BEV space based on the position of each vote, forming a sparse representation termed vanilla vote BEV features. Next, the non-empty grids of vanilla vote BEV are adaptively expanded to their neighboring regions according to the predicted class of the vote through a class-aware BEV diffusion block. Finally, we use a fully connected layer to project the class-aware vote BEV to the same channel dimension as the vanilla vote BEV, thus obtaining a diffused vote BEV feature.

output of the vote cluster module consists of two parts, *i.e.*, object-specific cluster features $\mathbf{F}_{clu}$ and their corresponding cluster centers $\mathbf{C}_{clu}$.

### 3.3 Feature-level Integration

In this subsection, we introduce the Cluster Feature Diffusion (CFD) module to generate diffused vote features in the unified BEV space, which are fused with the dense BEV features for representation enhancement. The unified BEV features are supervised by an imitation loss.

**Cluster Feature Diffusion.**  To establish a soft and adaptive association between cluster and BEV features for better fusing the two representations at the feature level, we propose the CFD module depicted in Figure 3, which involves two steps, *i.e.*, first converting the cluster features to the BEV space, and then using a class-aware BEV diffusion block to generate diffused cluster features. In the first step, given a series of vote centers $\mathbf{C}_{clu} = \{c_1, c_2, \cdots, c_N\} \in \mathbb{R}^{N \times 3}$ and vote cluster features $\mathbf{F}_{clu} = \{f_1, f_2, \cdots, f_N\} \in \mathbb{R}^{N \times C}$ from the cluster branch, we first map the voting centers to the BEV plane using the following Equation 1.

$$p_x = \frac{c_i^x - B_x}{v \cdot d}, \ p_y = \frac{c_i^y - B_y}{v \cdot d}, \ i = 1, 2, \cdots, N, \tag{1}$$

where $c_i^x$ is the coordinate of $x$-axis for the $i$-th vote center, $B_x$ is the boundary of $x$-axis for the given scene, $v$ is the voxel size of the BEV plane, and $d$ is the downsampling factor of the BEV feature map. Therefore, each cluster feature corresponds to one non-empty BEV pixel located at $p = (p_x, p_y)$, thus generating the vanilla vote BEV features (short for vote BEV). However, the established hard association between cluster and BEV features heavily relies on the quality of vote clusters. In fact, the cluster centers represent the potential locations of the centers of objects within a certain area.

To address this issue, in the second step, we propose the *class-aware BEV diffusion* block. Specifically, we first match the classification results from the cluster branch to the corresponding non-empty grids in the vanilla vote BEV. Secondly, we expand non-empty grids by diffusing their features into neighboring locations with simple max pooling operations on the BEV plane, where the diffusion factor varies in different vote classes to control the expansion magnitude. Thirdly, the generated class-aware vote BEV are together fed into a fully connected layer, which generates the required diffused vote BEV $\mathbf{F}_{voteB} \in \mathbb{R}^{C \times H \times W}$ as the output. A fully connected layer is used for channel-wise projection. Eventually, we can successfully convert the cluster features to the unified BEV plane in a soft and adaptive fashion, thus benefiting joint feature learning.

**Imitation Loss.**  Considering that the center feature of diffused vote BEV $\mathbf{F}_{voteB}$ and dense BEV $\mathbf{F}_{denseB}$ is not well aligned, the overall stability of representation learning may be hampered [5]. Therefore, to better distill object-centric knowledge from the cluster branch to the BEV branch and meanwhile avoid the imitation of areas of non-interest (*i.e.*, edge information of the object), we apply an imitation loss to the two features in the BEV space using Equations 2 and 3.

$$\mathcal{L}_{imi} = \text{Mask}\left(\mathbf{F}_{voteB}\right) \cdot \|\mathbf{F}_{voteB} - \mathbf{F}_{denseB}\|_2, \tag{2}$$

$$\text{Mask}\left(\mathbf{F}_{voteB}(p)\right) = \begin{cases} 0, \ if \ \mathbf{F}_{voteB}(p) \ is \ vaild, \\ 1, \ else, \end{cases} \quad (3)$$

where $\mathbf{F}_{voteB}$ and $\mathbf{F}_{denseB}$ denote the diffused vote BEV features from the CFD module and the dense BEV features from the BEV-based main branch. The Mask($\cdot$) operation only focuses on the certain center-oriented regions highlighted on the diffused vote BEV features, which indicates the potential areas of object centers.

### 3.4 Query-level Expansion

In this subsection, we introduce the Cluster Query Generation (CQG) module by expanding selected cluster queries which are implicitly supervised by a direction loss, thus enriching the object representation at the query level.

**Cluster Query Generation.** Most BEV detectors detect objects as center points. It is of significance to directly enrich the diversity of object queries by using the voting centers from the cluster branch. Therefore, we adopt a 3×3 convolution with sigmoid on the vanilla vote BEV, to generate a vote activation map. The location of the top $K$ activation scores will be taken out as cluster queries $\mathbf{Q}_{clu}$. The cluster queries are then directly concatenated with BEV queries $\mathbf{Q}_{BEV}$ (top $M$ candidates generated from BEV features). The positions and features of the hybrid queries $\{\mathbf{Q}_{BEV}; \mathbf{Q}_{clu}\}$ are used to initialize the query positions and query features of initial objects. The query embeddings are then fed into a transformer decoder to predict objects.

**Direction loss.** The quality of cluster queries is implicitly determined by the voting center of all the clusters. To ensure a more accurate voting center for each cluster, we employ a direction loss to constrain the direction of predicted offset vectors, which encourages all foreground points to point to their voting centers precisely. The direction loss [14] is defined based on the cosine similarities using the Equation 4.

$$\mathcal{L}_{dir} = -\frac{1}{\sum_i m_i} \cdot \frac{o_i}{\|o_i\|_2} \cdot \frac{b_i - p_i}{\|b_i - p_i\|_2} \cdot m_i, \quad (4)$$

where $m_i$ is the foreground mask from the segmentation head, $o_i$ is the offset vector, $p_i$ is the point coordinates and $b_i$ corresponds to the center of the ground-truth box in which the $p_i$ is located.

## 4 Experiments

In this section, we first make comparisons with the state-of-the-art methods on Waymo and nuScenes datasets. Then, we conduct ablation studies to examine the effect of each component of the proposed CluB.

**Waymo Open Dataset.** In Waymo Open Dataset [26] (WOD), 798, 202 and 150 sequences are used for training, validation and testing, respectively. We adopt the official evaluation metrics, *i.e.*, mean average precision (mAP) and mAPH (mAP weighted by heading) for Vehicle (Veh.), Pedestrian (Ped.), and Cyclist (Cyc.). The metrics are further split into two difficulty levels according to the number of points in ground truth boxes: LEVEL_1 (>5) and LEVEL_2 ($\geq$1).

**NuScenes Dataset.** NuScenes [2] dataset has 1,000 driving scenes, where 700, 150, and 150 scenes are chosen for training, validation and testing, respectively. The point cloud of nuScenes is collected by a 32-beam LiDAR. There are 10 classes in total. The evaluation metrics used by nuScenes are mAP and nuScenes detection score (NDS). The mAP is defined by the BEV center distance instead of the 3D IoU, and the final mAP is computed by averaging over distance thresholds of 0.5m, 1m, 2m, 4m across ten classes. NDS is a consolidated metric of mAP and other attribute metrics.

### 4.1 Implementation Details

**Training.** For Waymo, we follow previous voxel-based methods [22, 35, 23, 21] to use point cloud range of $[-75.2m, 75.2m] \times [-75.2m, 75.2m] \times [-2.0m, 4.0m]$ with voxel size $[0.1m, 0.1m, 0.15m]$ in $x$, $y$, and $z$-axes respectively. For nuScenes, we use point cloud range

| Methods | mAP/mAPH L2 | Vehicle 3D AP/APH L2 | L1 | Pedestrian 3D AP/APH L2 | L1 | Cyclist 3D AP/APH L2 | L1 |
|---|---|---|---|---|---|---|---|
| IA-SSD△ [36] | 62.3/58.1 | 61.6/61.0 | 70.5/69.1 | 60.3/50.7 | 69.4/58.5 | 65.0/62.7 | 67.7/65.3 |
| FSD△† [10] | 72.9/70.8 | 70.5/70.1 | **79.2/78.8** | 73.9/ 69.1 | **82.6/77.3** | 74.4/73.3 | 77.1/76.0 |
| CenterPoint [35] | -/67.4 | -/67.9 | -/- | -/65.6 | -/- | -/68.6 | -/- |
| PV-RCNN [22] | 66.8/63.3 | 69.0/68.4 | 77.5/76.9 | 66.0/57.6 | 75.0/65.6 | 65.4/64.0 | 67.8/66.4 |
| AFDetV2 [12] | 71.0/68.8 | 69.7/69.2 | 77.6/77.1 | 72.2/67.0 | 80.2/74.6 | 71.0/70.1 | 73.7/72.7 |
| SST_TS [9] | -/- | 68.0/67.6 | 76.2/75.8 | 72.8/65.9 | 81.4/74.1 | -/- | -/- |
| PillarNet-34 [21] | 71.0/68.5 | **70.9/70.5** | 79.1/**78.6** | 72.3/66.2 | 80.6/74.0 | 69.7/68.7 | 72.3/71.2 |
| PV-RCNN++ [23] | 71.7/69.5 | 70.6/**70.2** | 79.3/78.8 | 73.2/68.0 | 81.3/76.3 | 71.2/70.2 | 73.7/72.7 |
| TransFusion-L [1] | -/64.9 | -/65.1 | -/- | -/63.7 | -/- | -/65.9 | -/- |
| CenterFormer [38] | 71.2/69.0 | 70.2/69.7 | 75.2/74.7 | 73.6/68.3 | 78.6/73.0 | 69.8/68.8 | 72.3/71.3 |
| ConQueR [39] | 70.3/67.7 | 68.7/68.2 | 76.1/75.6 | 70.9/64.7 | 79.0/72.3 | 71.4/70.1 | 73.9/72.5 |
| ConQueR⋆ [39] | **74.0/71.6** | **71.0/70.5** | 78.4/77.9 | **75.8/70.1** | 82.4/76.6 | **75.2/74.1** | 77.5/76.4 |
| VoxelNext [4] | 70.9/68.2 | 69.7/69.2 | 77.9/77.5 | 72.2/65.9 | 80.2/73.5 | 70.7/69.6 | 73.3/72.2 |
| VoxelNext-K3‡ [4] | 72.2/70.1 | 69.9/69.4 | 78.2/77.7 | 73.5/68.6 | 81.5/76.3 | 73.3/72.2 | 76.1/74.9 |
| CluB$_{light}$ (ours) | 71.2/69.3 | 68.4/68.0 | 76.5/76.0 | 71.1/66.1 | 79.4/73.9 | 74.3/73.0 | 76.7/75.3 |
| CluB (ours) | **73.4/71.4** | 70.4/69.9 | 79.1/**78.6** | **74.3/69.5** | **83.1/77.9** | **75.6/74.9** | **78.2/77.5** |

Table 1: Performances on the WOD *validation* split. All models take single-frame input, no pre-training or ensembling is required. $_{light}$ denotes a light version that involves simple feature aggregation without further feature extraction for cluster features [19]. † denotes using a longer input range for the point cloud. ‡ denotes using larger model size (*e.g.*, 5× kernel size for 3D sparse encoder) for enhancement. ⋆ denotes an enhancement version (*i.e.*, wider backbone network and additional NMS for Pedestrian and Cyclist). △ denotes that belonging to the 3D detector featured by cluster-based paradigm. We highlight the top-2 entries with **bold** font.

of $[-51.2m, 51.2m] \times [-51.2m, 51.2m] \times [-5.0m, 3.0m]$ with voxel size $[0.1m, 0.1m, 0.2m]$ in $x$, $y$, and $z$-axes respectively. We adopt the same data augmentation setting as [35], including random flipping, global scaling, global rotation, and groundtruth (GT) sampling [33] for Waymo dataset. These settings are similar to those of nuScenes dataset following [1]. We use the one-cycle [25] learning rate schedule and AdamW [16] optimizer with the maximal learning rate 0.001. In addition to the 12-epoch schedule for ablation studies, we adopt a longer schedule (36 epochs) to obtain the best performance on the *validation* and *test* set. During the evaluation, we use the NMS IoU threshold of [0.7, 0.25, 0.25]. Our code is built on MMdetection3D [6].

**Network.** The 3D sparse encoder and decoder are built upon sparse U-Net in PartA2 [24]. For the BEV backbone, we use the same architecture as SECOND [33] and obtain multi-scale BEV features with FPN structure. The top 300 BEV queries are selected from the predicted heatmap following [1] The vote cluster module generates cluster features with 128 dimensions via simple aggregation [19] or further instance-wise feature extraction [10]. For the CFD module, we set the diffusion factors according to the size of different classes. For example, we set 1, 3, and 5 for pedestrians, cyclists, and vehicles in Waymo. For the CQG module. we generate top-ranking cluster queries from the activation map. Since the maximum number of objects in one frame is 185 and 142 for Waymo and nuScenes datasets [1], the number of cluster queries is set as 200 and 150, respectively. We set the number of transformer layers and heads to 3 and 8, and the category labels are simply encoded as one-hot embeddings following [1].

## 4.2 Main Results

**Waymo Results.** We summarize the performance of CluB and state-of-the-art (SOTA) 3D detection methods on WOD *validation* and *test* sets in Table 1 and Table 2. As shown in Table 1, our CluB achieves competitive single-frame performance and sets new records on *Cyclist* and *Pedestrian* of the WOD *validation* set. CluB achieves comparable mAPH/L2 performance on the *validation* set compared with the previous best transformer-based model ConQueR [39].

| Methods | All | Veh. | Ped. | Cyc. |
|---|---|---|---|---|
| CenterPoint [35] | 69.0 | 71.9 | 67.0 | 68.2 |
| PV-RCNN++ [23] | 70.2 | 73.5 | 69.0 | 68.2 |
| AFDetv2 [12] | 70.0 | 72.6 | 68.6 | 68.7 |
| PillarNet-34 [21] | 69.6 | **74.7** | 68.5 | 65.5 |
| ConQueR [39] | 72.0 | 73.3 | **70.9** | 71.9 |
| CluB (Ours) | **72.2** | 73.4 | 70.8 | **72.2** |

Table 2: Performance comparison of different single-frame models on the WOD *test* set. APH/L2 results are reported.

| Methods | NDS | mAP | Car | Truck | Bus | Trailer | C.V. | Ped. | Motor. | Bicyc. | T.C. | Barrier |
|---|---|---|---|---|---|---|---|---|---|---|---|---|
| AFDetV2 [12] | 68.5 | 62.4 | 86.3 | 54.2 | 62.5 | 58.9 | 26.7 | 85.8 | 63.8 | 34.3 | 80.1 | 71.0 |
| CenterPoint [35] | 67.3 | 60.3 | 85.2 | 53.5 | 63.6 | 56.0 | 20.0 | 84.6 | 59.5 | 30.7 | 78.4 | 71.1 |
| TransFusion-L [1] | 70.2 | 65.5 | 86.2 | 56.7 | 66.3 | 58.8 | 28.2 | 86.1 | 68.3 | 44.2 | 82.0 | **78.2** |
| VoxelNext [4] | 70.0 | 64.5 | 84.6 | 53.0 | 64.7 | 55.8 | 28.7 | 85.8 | **73.2** | **45.7** | 79.0 | 74.6 |
| CluB (Ours) | **71.2** | **66.0** | **87.2** | **57.0** | **66.4** | **59.0** | **28.8** | **87.2** | 69.0 | 45.4 | **82.1** | 78.2 |

Table 3: Performance comparison of different models on the nuScenes dataset.

Meanwhile, CluB achieves 0.6% mAPH/L2 performance gain than the powerful detector FSD [10] featured by cluster-based paradigm, even FSD takes longer-range points as input. CluB surpasses the BEV-based method TransFusion-L, which is also our BEV-based baseline, by **6.5%** mAPH/L2. It is noteworthy that CluB achieves 1.3% mAPH/L2 performance improvement compared to Voxel-Next [4] which utilizes a larger model size for model enhancement(*e.g.,* $5\times$ kernel size for 3D sparse encoder). Furthermore, with the help of further cluster-wise feature extraction, CluB improves the detection performance of the light version $CluB_{light}$ on all classes. Notably, $CluB_{light}$ consistently demonstrates competitive performance on the *Cyclist class*, compared with all previous single-frame methods in terms of both APH/L1 and APH/L2.

In addition to offline results, we also report the detection performance on Waymo *test* set to explore the potential of CluB. As shown in Table 2, our CluB outperforms the previous single-frame LiDAR-only methods on the *test* set, *e.g.*, 2.0% / 0.2% mAPH/L2 improvement compared with PV-RCNN++ / ConQueR, respectively.

Here, we also give our analysis of why CluB performs better on the cyclist and pedestrian classes. Actually, the cluster-based branch works better when the point clouds are more compact, which means the improvements are more significant for cyclists than that for vehicles. Due to the fixed voxelization and over-downsampling, it is challenging to capture sufficient details of small objects with the BEV-based branch. On the contrary, the cluster branch becomes highly beneficial to extracting fine-grained features from sparser points with smaller cluster voxel sizes, which leads to notable performance improvement.

**NuScenes Results.** We summarize the performance of CluB and different baselines on the nuScenes *test* set in Table 3. Notably, compared with the concurrent fully sparse detector VoxelNext [4], CluB achieves **1.2%** / **1.5%** improvement on NDS / mAP (70.0 → 71.2 , 64.5 → 66.0). Compared with the strong BEV-based baseline Transfusion-L, CluB also leads to **1.0**% and **0.5**% improvement on NDS and mAP (70.2 → 71.2, 65.5 → 66.0 ). The experimental results on Waymo and nuScenes benchmarks validate the effectiveness of CluB by integrating the cluster representation to the BEV branch via the two-level enhancement scheme.

## 4.3 Ablation Study

To validate the effect of each component in CluB, we conduct ablation studies on the WOD. The models compared in this section are trained with 20% training samples of WOD, and evaluated on the whole validation set.

**Effect of Components in CluB.** To understand how each module contributed to the final performance in CluB, we test each component independently and report its performance in Table 4. Our baseline detector (a) employs the mainstream BEV paradigm, which starts from 60.5% mAPH/L2. When the CFD module is applied, the mAPH/L2 of *Method (b)* is raised by 1.4%, which indicates that adaptively integrating the cluster feature in a soft manner is effective to improve 3D detection. *Method (c)* brings a 2.0% mAPH/L2 improvement by introducing an imitation loss, which validates that explicitly transferring knowledge (object-specific features) to the BEV stream is necessary. On the other hand, *Method (d)* adds the CQG module based on the *Method (a)* and brings a total of 0.5% mAPH/L2 enhancement by taking advantage of querying expansion. *Method (e)* further introduces a direction loss to ensure the quality of cluster queries with 1.1 improvements. The combination of all the components in *Method (f)* achieves 63.9% mAPH/L2 (3.4 % absolute improvement), validating the effectiveness of CluB. This indicates that enhancing the object representation from two branches at the feature and query levels is of great use to improve detection accuracy.

| Method | Feature-level | | Query-level | | mAPH/L2 |
|---|---|---|---|---|---|
| | CFD Module | Imitation Loss | CQG Module | Direction Loss | |
| (a) | | | | | 60.5 |
| (b) | ✓ | | | | 61.9 ↑ 1.4 |
| (c) | ✓ | ✓ | | | 62.5 ↑ 2.0 |
| (d) | | | ✓ | | 61.0 ↑ 0.5 |
| (e) | | | ✓ | ✓ | 61.6 ↑ 1.1 |
| (f) | ✓ | ✓ | ✓ | ✓ | 63.9 ↑ 3.4 |

Table 4: Effect of each component in CluB. This experiment reveals that the combined object representation is indeed strengthened by elaborately fusing the context-aware BEV representation and object-centric cluster representation at both the feature level and the query level.

| Strategy | Operation | Using diffusion | Using class infor. | APH/L2 | | |
|---|---|---|---|---|---|---|
| | | | | Veh. | Ped. | Cyc. |
| (a) | Direct concatenation | | | 61.9 | 63.0 | 63.9 |
| (b) | 2D convolution | ✓ | | 61.4↓ 0.5 | 62.5↓ 0.5 | 63.7↓ 0.4 |
| (c) | Max pooling | ✓ | | 61.0↓ 0.9 | 63.4↑ 0.4 | 64.1↑ 0.2 |
| (d) | Max pooling | ✓ | ✓ | 62.9↑ 1.0 | 64.4↑ 1.4 | 64.5↑ 0.6 |

Table 5: Effect of different strategies for the CFD module. This experiment reveals the importance of establishing the association between cluster and BEV features in a soft and adaptive way when enhancing object representation at the feature level.

**Effect of Diffusion Strategies in the CFD Module.** We compare our proposed class-aware BEV diffusion namely *Strategy (d)* with other strategies, which are shown in Table 5. Intuitively, we can directly concatenate the cluster feature with the BEV feature based on the location of the voting centers, which is a sub-optimal strategy discussed in Section 1. We consider the naive *Strategy (a)* relying on hard association as the baseline strategy in Table 5. *Strategy (b)* uses a 2D convolution layer to diffuse valid features to the neighboring, which not only introduces extra parameters but degrades the overall detection performance. In contrast, *Strategy (c)* utilizes the max pooling operation, an elegant solution to avoid some unnecessary computation for a more effective representation learning procession. Compared with the class-agnostic manner (c), our diffusion scheme (d) leads to better performance, *e.g.*, 61.0 → 62.9, 63.4 → 64.4 APH/L2 on vehicle and pedestrian by considering semantic consistency. *Strategy (d)* establishes the soft and adaptive association between cluster and BEV features, thus successfully enhancing object representation at the feature level.

**Effect of Design Choices of Cluster Module.** We examine the effects of different design choices of cluster modules in Table 6. The base competitor *Method (a)* is a BEV-based 3D detector we implemented. Compared with CluB, it abandons the cluster branch. As evidenced in the *Method (a)* and the *Method (b)*, by integrating the cluster branch via a simple aggregation strategy, *i.e.*, K-Nearest Neighbor (KNN), *Method (b)* performs better, *e.g.*, 61.0 → 61.4, 60.4 → 64.3 on vehicle and pedestrian. Combining more local features for each cluster (8 neighbors of KNN), our *Method (c)* achieves a significant improvement on vehicle class, *i.e.*, 61.0 → 63.2. However, for the pedestrian class, the performance degrades from 60.4 to 59.9 because points that do not belong to pedestrians are aggregated. That indicates that a well-designed clustering algorithm can improve the overall performance. Motivated by that, *Method (d)* replaces the KNN algorithm with the ball query algorithm, and the detection performance steadily improves among the three classes. *Method (e)* utilize the elaborate cluster module following [10], which groups points into instance-wise clusters, achieving the satisfactory detection performance shown in the last row in Table 6.

## 5 Conclusion

In this paper, we introduce CluB, a unified 3D object detection framework that takes advantage of both BEV-based and cluster-based paradigms for improving the accuracy of 3D object detection. To effectively combine the context-aware BEV representation and object-centric cluster representation,

| Methods | Cluster Branch | Cluster Module Design | APH/L2 | | |
|---|---|---|---|---|---|
| | | | Veh. | Ped. | Cyc. |
| (a) | | | 61.0 | 60.4 | 60.1 |
| (b) | ✓ | KNN-4 | 61.4 | 64.3 | 64.1 |
| (c) | ✓ | KNN-8 | **63.2** | 59.9 | 64.4 |
| (d) | ✓ | Ball query | 62.7 | 63.8 | 64.4 |
| (e) | ✓ | SIR | 62.9 | **64.4** | **64.5** |

Table 6: Effects on different design choices of vote cluster modules. The experimental result reveals that integrating the cluster branch is beneficial for detection performance, and a more elaborately designed grouping strategy may bring more gains.

we propose to integrate an auxiliary cluster-based branch to the mainstream BEV-based detector by strengthening the object representation at both feature and query levels. On the one hand, we first establish the association between cluster and BEV features in a soft adaptive way using the CFD module. Based on the association, we transfer the object-specific knowledge from the cluster to the BEV branch supervised by an imitation loss. On the other hand, we design a CQG module to enrich the diversity of object queries using the voting centers directly from the cluster branch. Simultaneously, a direction loss is employed to encourage a more accurate voting center for each cluster. The two-level enhancement (*i.e.*, feature integration, and query expansion) design of CluB greatly enriches the object representation ability. With our proposed framework, CluB outperforms the previous single-frame models for detecting 3D objects on both Waymo and nuScenes datasets.

**Limitations.**  Remarkably, CluB presents an elegant view of how to combine the context-aware BEV representation with object-centric cluster representation in a unified detection framework and achieves promising performance via two-level enhancement (feature, query) for object representation. Yet, the computational and spatial complexity on BEV feature maps is quadratic to the perception range, which makes it hard to adopt our CluB for long-range detection. We leave the exploration of computational efficient version of our method as the future work.

# 6  Acknowledgement

We hereby thank Pengfei Luo for his contribution. This work was supported by the Chinese Academy of Sciences Frontier Science Key Research Project ZDBS-LY-JSC001, the National Key R&D Program of China (No.2022ZD0160100), and in part by Shanghai Committee of Science and Technology (No.21DZ1100100).

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
