## A    Training Objectives

Our model is trained from scratch with the semantic loss $\mathcal{L}_{sem}$, voting loss $\mathcal{L}_{vote}$ [3], direction loss $\mathcal{L}_{dir}$ for the cluster auxiliary branch. In terms of the BEV main branch, we follow the same loss functions $\mathcal{L}_{cls}$, $\mathcal{L}_{box}$ and $\mathcal{L}_{heatmap}$ defined in [1] to optimize object classification, box regression and heatmap prediction, respectively. Note that the $\mathcal{L}_{heatmap}$ is computed both from two branches. Meanwhile, imitation loss $\mathcal{L}_{imi}$ is computed. The total loss $\mathcal{L}_{total}$ is the weighted sum of losses for each component, which is presented in Equation 1.

$$\mathcal{L}_{total} = \beta_{sem}\mathcal{L}_{sem} + \beta_{vote}\mathcal{L}_{vote} + \beta_{dir}\mathcal{L}_{dir} + \beta_{cls}\mathcal{L}_{cls} + \beta_{box}\mathcal{L}_{box} + \beta_{heatmap}\mathcal{L}_{heatmap} + \beta_{imi}\mathcal{L}_{imi} \tag{1}$$

## B    Model Efficiency

We present the comparison of model efficiency between our CluB and the BEV-only baseline in terms of latency and FLOPs accordingly. The results are obtained on Waymo Open Dataset with one NVIDIA 3090 GPU. The computational overhead of CluB is 1.2 / 1.3 times that of the BEV-only baseline in terms of latency / FLOPs. The cost is affordable since the auxiliary cluster branch is built with fully sparse operations[2]. A detailed comparison is shown in the following table.

|              | Baseline | CluB   |
| ------------ | -------- | ------ |
| Latency (ms) | 144      | 167    |
| FLOPs (G)    | 95.71    | 112.35 |

Table 1: Comparison between our CluB and baseline on the model efficiency.

## C    More Implementation Details

Following [4], we adopt a UNet-like architecture for learning point-wise feature representations with 3D sparse convolution and 3D sparse deconvolution on the obtained sparse voxels. The spatial resolution is downsampled 8 times by three sparse convolutions of stride 2, each of which is followed by several submanifold sparse convolutions. For the decoder, there are four sparse up-sampling blocks to gradually increase the spatial resolution. Note that the stride of the last up-sampling block is 1 and the stride of other three up-sampling blocks is 2.

For training, we use the fade strategy proposed in [5] to drop GT-Aug at the last epoch for nuScenes and the last three epochs for Waymo to avoid overfitting. We train our model on 8 NVIDIA A100 GPUs and the batch size per GPU is set as 2.

## D    Effect of Two Kinds of Object Queries for the Transformer Decoder

Compared to CluB (the first row in Table 2), if we remove the cluster queries (the second row), we observe a significant drop in the performance of cyclist and pedestrian classes. This decline suggests that cluster queries play a beneficial role in enhancing the diversity of object queries, particularly for fine-grained targets. By removing the BEV queries alone (the last entry), overall performance drops over 6.3% mAPH/L2. We conclude that BEV queries directly indicate the possible positions of objects on the BEV space, making it easier for the decoder to detect target objects. Therefore, BEV queries and cluster queries have their respective advantages in query initialization for the decoder, thus enhancing detection performance.

## E    Qualitative Results

We provide the visual comparison between the BEV-only 3D detector TransFusion-L [1] and our proposed CluB in Figure 1. Specifically, blue circles represent the inaccurate detections of the baseline but CluB corrects them by integrating the cluster branch.

| BEV Queries | Cluster queries | APH/L2 | | | mAPH/L2 |
|:---:|:---:|:---:|:---:|:---:|:---:|
| | | Veh. | Ped. | Cyc. | |
| ✓ | ✓ | 62.9 | 64.4 | 64.5 | 63.9 |
| ✓ | | 62.6 ↓ 0.3 | 62.7 ↓ 1.7 | 62.0 ↓ 2.5 | 62.5 ↓ 1.4 |
| | ✓ | 56.9 ↓ 6.0 | 57.7 ↓ 6.7 | 58.1 ↓ 6.4 | 57.6 ↓ 6.3 |

Table 2: Ablation study on the effect of the two kinds of object queries for the transformer decoder.

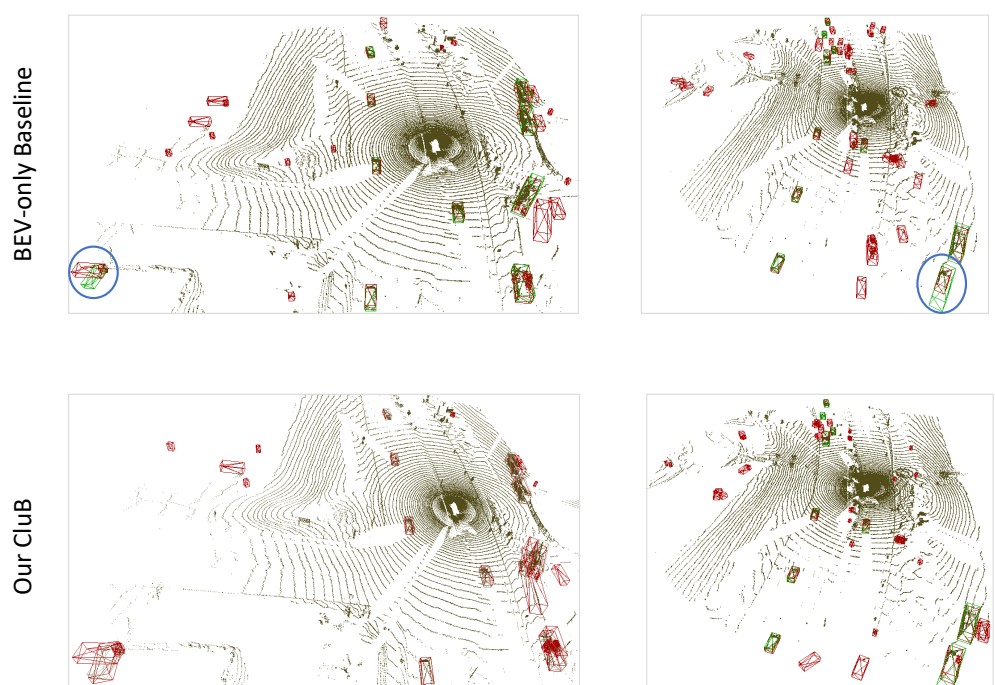

Figure 1: Qualitative comparison between the BEV-only 3D detector TransFusion-L [1] and our CluB on Waymo dataset. For all classes, we reserve the 20 boxes with the highest scores for visualization after the NMS. Red boxes and green boxes are the predictions and ground-truth, respectively. Best viewed with color and zoom-in.