# OpenReview forum: "CluB: Cluster Meets BEV for LiDAR-Based 3D Object Detection"
_NeurIPS.cc/2023/Conference — NeurIPS 2023 poster_

### Official Review · Reviewer_UKd9 · 2023-06-20

**Soundness:** 3 good
**Presentation:** 3 good
**Contribution:** 3 good
**Rating:** 6
**Confidence:** 5

**Summary:**

This paper combines two streams of LiDAR-based 3D object detectors into a single model: conventional BEV-based detectors and emerging cluster-based detectors. The main contribution is that It is the first one to combine the two streams of methods, leveraging the high-recall of BEV representation and fine-grained structure of cluster representation. It reveals the potential of such a combination, providing a possibility for a new sub-direction.


**Strengths:**

- This paper is well written with clear representation and concise illustration. For example, Figure 2 is very informative and concise, and I could clearly get the overall idea at first glance.

- To my best knowledge, it is the first one to explicitly combine the two streams of methods, leveraging the high-recall of BEV representation and fine-grained structure of cluster representation. Although straightforward, the idea is well-motivated and makes sense to me.

- Every module is well ablated. Overall performance on Waymo Open Dataset and nuScenes dataset is good. Especially, on nuScenes, it achieves state-of-the-art performance.

**Weaknesses:**

- Although the overall writing is clear, the authors are not very careful with some details. For example, Figure 1 (a) is very similar to a figure in the FSD paper, and it would be appropriate to add a citation. In L108, the citations should be placed after "benchmarks" instead of detectors. In Eqn (4), $Ldir$ should be $L_{dir}$.  There are many errors in Table 1: the best and second-best entries of L2 Vehicle should be PillarNet and PVRCNN++. The performance of FSD seems inaccurate because it has a similar vehicle performance to PVRCNN++ if I recall correctly. Please check. The second-best Pedestrian L2 APH is not underlined. And this table size is out of the range on the right.

- Although experiments thoroughly verify the effectiveness of proposed modules, they are straightforward ablation without insight. I encourage authors to add more detailed performance analysis to reveal the inner workings of CluB. For example, how do the two kinds of queries affect the final performance?  Since BEV-based representation has higher recall and cluster-based representation preserves fine-grained structure, authors should design experiments to demonstrate such properties, which cannot be demonstrated by simple numbers of the final performance. Table 1 in supplementary materials should be moved to the main paper, and the current 9-page space is not fully utilized.

- I encourage authors to conduct runtime evaluations and report the latency.

**Questions:**

I like the idea of combining the two kinds of representation. However, the authors should carefully address my concerns.
The manuscript should be updated in the OpenReview system if possible to address the first weakness. Additional experiments could be simply posted on this page for now.

**Limitations:**

The authors adequately addressed the limitations.

---

> ### Author Rebuttal · Authors · 2023-08-10
>
> We sincerely appreciate your constructive and detailed comments. We respond to your comments below.
>
> > *detail errors and format correct*
>
> Thanks for pointing out these typos. We have addressed all the errors you mentioned, and we will proofread them very carefully. Since the manuscript can not be updated in the OpenReview system at present, we will make sure the fore-mentioned errors are corrected in the final manuscript.
>
> > *Errors of Table 1.*
>
> Thanks for pointing it out. We make the corresponding corrections to address the problems.
> - We checked the performance of FSD and modified the numbers in the revised Table 1. The best vehicle  performance in the FSD (70.1%) paper is indeed similar to PVRCNN++ (70.2%) in terms of APH/L2.
> - As FSD adopts a longer point cloud range, we apply the same input range on our CluB to further enhance the performance (71.4% mAPH/L2). The enhanced CluB is trained on 8 NVIDIA RTX A6000 GPUs with larger batch size of 4. The latest result is also reported on the revised Table 1.
> - We have carefully proofread the label of the best and second-best entries and revised the table size as pointed out.
>
> Since the manuscript can not be updated in the OpenReview system at present, we present **part of the revised Table 1** (Table 5 in the PDF). Thank you again for your detailed thorough review.
>
>
> > *Although experiments thoroughly verify the effectiveness of proposed modules, they are straightforward ablation without insight. I encourage authors to add more detailed performance analysis to reveal the inner workings of CluB. For example, how do the two kinds of queries affect the final performance? Since BEV-based representation has higher recall and cluster-based representation preserves fine-grained structure, authors should design experiments to demonstrate such properties, which cannot be demonstrated by simple numbers of the final performance.*
>
> Thanks for your insightful comment. Based on your suggestion, we conduct both quantitative and qualitative experiments to show how the two cluster queries and BEV queries affect the final performance. We first provide the quantitative results on the Waymo dataset in the table below (Table 4 in the PDF).
>
> Compared to CluB (the first row), if we remove the cluster queries (the second row), we observe a significant drop in the performance of cyclist and pedestrian classes. This decline suggests that cluster queries play a beneficial role in enhancing the diversity of object queries, particularly for fine-grained targets.  By removing the BEV queries alone (the last entry),  overall performance drops over 6.3% mAPH/L2. We conclude that BEV queries directly indicate the possible positions of objects on the BEV space, making it easier for the decoder to detect target objects. Therefore, BEV queries and cluster queries have their respective advantages in query initialization for the decoder, thus enhancing detection performance.
>
> To intuitively demonstrate the contribution of these two queries, we also visualize the two kinds of queries on the same BEV features using Waymo dataset, which can be seen in the Figure 3 of the PDF. The yellow square represents the BEV query activated by the BEV heatmap, while the pink cross signifies the cluster query derived from vote center positions in 3D space. The red circle denotes a potential object initialization position overlooked by the BEV query. This comparison reveals that BEV queries provide a more comprehensive coverage of potential object locations, whereas cluster queries enrich object diversity with finer granularity.
>
> We will add the experimental results and analysis in the future revision. Thanks again for your advice.
>
>
> | BEV queries | Cluster queries | Vehicle | Pedestrian | Cyclist | mAPH/L2 |
> |-------------|-----------------|---------|------------|---------|---------|
> | √           | √               | 62.9    | 64.4       | 64.5    | 63.9    |
> | √           |                 | 62.6 ↓0.3    | 62.7 ↓1.7      | 62.0 ↓2.5     | 62.5 ↓1.4   |
> |             | √               | 56.9 ↓6.0   | 57.7 ↓ 6.7      | 58.1 ↓6.4   | 57.6 ↓6.3   |
>
> > *I encourage authors to conduct runtime evaluations and report the latency.*
>
> Thank you for the suggestion. We evaluate the runtime on single NVIDIA GEFORCE RTX 3090. The computational overhead of CluB is 1.2 times that of the BEV-only baseline in terms of latency. Please see the table below:
>
> |              | Baseline | CluB |
> |--------------|----------|------|
> | Latency (ms) | 144      | 167  |

---

> > ### Comment · Reviewer_UKd9 · 2023-08-10
> > **Authors have addressed most of my concerns.**
> >
> > Thanks for your feedback and I am glad to read such an informative rebuttal. Fig. 3 is exactly what I suggested, and I believe more analysis can be done to gain deeper insight (I am not asking for them now). Before my final decisions, I have a few more concerns:
> > - Although FSD uses a longer range, I do not think a longer range could lead to notable improvements in Waymo because there are very few objects beyond 75 meters. Are you sure that your performance improvement of the enhanced version is from adopting a longer range?
> > - When I refer to the original FSD paper, I found you cited the wrong paper (CenterFormer) in Table 1.
> >
> > No more experiments are needed, just discussion.

---

> > > ### Author Response · Authors · 2023-08-11
> > > **Response to Reviewer UKd9**
> > >
> > > We sincerely appreciate the suggestions and comments. Your insightful comments have greatly enhanced the quality of this paper. We address the concerns and questions you raised as follows.
> > >
> > > >*Although FSD uses a longer range, I do not think a longer range could lead to notable improvements in Waymo because there are very few objects beyond 75 meters. Are you sure that your performance improvement of the enhanced version is from adopting a longer range?*
> > >
> > > Thanks for your valuable comments. In fact, we also increased the batch size from 2 to 4 and trained the enhanced model on 8 A6000 GPUs as mentioned in the previous response. These factors may also contribute to the improved performance. We will further attempt to pinpoint the exact reason and provide a more accurate explanation in the camera-ready version.
> > >
> > > >*When I refer to the original FSD paper, I found you cited the wrong paper (CenterFormer) in Table 1.*
> > >
> > > Thank you for kindly pointing it out. We will proofread Table 1 very carefully and make sure the aforementioned error is corrected in the final version.
> > >
> > > Please kindly let us know if there are any other issues that have not been fully addressed. We would be more than happy to engage in further discussion to clarify them. Thank you for your time and attention.

---

> > > > ### Comment · Reviewer_UKd9 · 2023-08-12
> > > > **Technical solid paper, no more major concerns, worthy of acceptance.**
> > > >
> > > > After reading the responses to all reviewers and engaging in further discussion, the authors have addressed most of my concerns.
> > > > I believe it is a technically solid paper and a good improvement on the cluster-based detectors.
> > > > So I increase my rating to 6.
> > > > Authors are supposed to add the additional experiments into the next version and I look forward to the next version with better performance/analysis/writing.

---

### Official Review · Reviewer_KK49 · 2023-07-04

**Soundness:** 2 fair
**Presentation:** 2 fair
**Contribution:** 2 fair
**Rating:** 5
**Confidence:** 4

**Summary:**

This work combines the BEV-based and cluster-based representations into a unified framework named CluB. At the feature level, the Cluster Feature Diffusion module and the imitation loss are proposed to fuse the features obtained from the BEV branch and cluster branch. At the query level, the Cluster Query Generation module and the direction loss are proposed to provide more accurate object queries from the cluster branch. CluB achieves state-of-the-art performance on the Waymo and nuScenes datasets.

**Strengths:**

This work mentions that the features of the object centers in BEV-based detectors are diffused from their neighbors and are not in high quality, which is an interesting hypothesis. Based on that, the cluster branch is applied to provide clustered features of the object centers. The whole structure of the work is compact and unified, and the promotion on precision is clearly shown.

**Weaknesses:**

1. The argumentation of the core hypothesis is insufficient. I think that the max pooling in BEV-based detectors is able to reduce the feature shape and aggregate the feature of the object to the center, just like the cluster-based methods. Additional quantitative analysis is preferred to show the difference.
2. The captions of figures and tables are too long and mainly repeat the text of the main body.

**Questions:**

1. Some results in the ablation study do not match with each other. For example, the mAPH of Table 4 (b) should be the same as (d), but in fact they are not. Are there differences between the two configurations?
2. I am wondering why CluB performs better on the cyclist class of Waymo, because the point clouds of cyclists are more compact than the vehicles and should benefit less from the cluster branch.
3. How efficient is CluB compared to the baseline? The time cost could be significantly increased by applying U-Net in the cluster branch.

**Limitations:**

It would be nice to apply different BEV-based detectors as the baseline to test the universality of the method.

---

> ### Author Rebuttal · Authors · 2023-08-10
>
> We sincerely appreciate your constructive and detailed comments.
>
> > *The argumentation of the core hypothesis is insufficient. I think that the max pooling in BEV-based detectors is able to reduce the feature shape and aggregate the feature of the object to the center, just like the cluster-based methods. Additional quantitative analysis is preferred to show the difference.*
>
> Thanks for your thought-provoking comment.  Accordingly, we conducted an additional experiment that quantitatively verifies max pooling in BEV-based detectors is not able to work like the cluster-based method. The results are summarized as the following table (Table 2 in the PDF).
> | Method           | Vehicle | Pedestrian | Cyclist | mAPH/L2 |
> |------------------|---------|------------|---------|---------|
> | Baseline         | 61.0     | 60.4       | 60.1    | 60.5    |
> | Baseline_maxpool | 60.5    | 59.9       | 59.1    | 60.0      |
> | Baseline_CFD     | **61.9**    | **61.2**       | **62.5**    | **61.9**    |
>
>
> Specifically, Baseline_maxpool takes sparse voxels directly from the sparse U-Net and transforms them into a dense BEV feature. Subsequently, several pre-defined max pooling layers are applied to amplify the object's feature towards its center. In contrast, Baseline_CFD replaces the multiple max-pooling layers with our cluster feature diffusion (CFD) module. This module dynamically merges cluster features with BEV features to enhance the center's feature.
>
> As demonstrated in the table, the mAPH/L2 is dropped from 60.5% to 60.0% for Baseline_maxpool, which demonstrates that max pooling operations on BEV features cannot effectively aggregate features to the center of the target to enhance the representation of target features. In comparison, the cluster representation with its adaptable receptive fields demonstrates superior effectiveness in achieving this enhancement. We will include the experimental results and analysis in the future revision. Thanks again for your advice.
>
> > *The captions of figures and tables are too long and mainly repeat the text of the main body.*
>
> Thanks for your kind advice. We will shorten the captions of figures and tables in the revised manuscript  as suggested.
>
> > *The mAPH of Table 4 (b) should be the same as (d), but in fact they are not. Are there differences between the two configurations?*
>
> Thanks for the comment. The configurations of methods (b) and (d) in Table 4 are in fact different. Compared to the baseline, Method (b) adds the CFD module (feature-level) while Method (d) introduces the CQG module (query-level).
>
> Specifically, Method (b) adds the Cluster Feature Diffusion (CFD) module compared to the baseline BEV-based detector, which adaptively generates diffused vote features in the unified BEV space for the feature representation enhancement. Method (d) only introduces the Cluster Query Generation (CQG) module compared to the baseline, which enriches the diversity of object queries by using the positions of voting centers from the cluster branch.
>
> > *I am wondering why CluB performs better on the cyclist class of Waymo, because the point clouds of cyclists are more compact than the vehicles and should benefit less from the cluster branch.*
>
> Actually, the cluster-based branch works better when the point clouds are more compact, which means the improvements are more significant for cyclists than that for vehicles. Due to the fixed voxelization and over-downsampling, it is challenging to capture sufficient details of small objects with the BEV-based branch. On the contrary, the cluster branch becomes highly beneficial to extract fine-grained features from sparser points with smaller cluster voxel size, which leads to notably performance improvement.
>
> > *How efficient is CluB compared to the baseline? The time cost could be significantly increased by applying U-Net in the cluster branch.*
>
> Thank you for the suggestion. The computational overhead of CluB is 1.2 / 1.3  times that of the BEV-only baseline in terms of latency / FLOPs respectively. We present the efficiency-related statistics of our CluB and baseline (w.o. cluster branch) on Waymo dataset below:
> |              | Baseline | CluB   |
> |--------------|----------|--------|
> | Latency (ms) | 144      | 167    |
> | FLOPs (G)    | 95.71    | 112.35 |
>
> The time cost is not significantly increased for the following reasons. First, the decoder of the U-Net is shared between both the BEV-based baseline and our approach to extract 3D voxel features. Besides, the cluster branch primarily involves fully sparse operations [1] on the given point set. The latency is evaluated using one NVIDIA GEFORCE RTX 3090 GPU.
>
> > *It would be nice to apply different BEV-based detectors as the baseline to test the universality of the method.*
>
> Thanks for the valuable suggestion. We choose CenterPoint [2], which is also a prevalent BEV baseline in the community, to test the universality of the method. We provide the results in the following table(Table 3 in the PDF).
>
> CenterPoint is an anchor-free one-stage detector, which extracts BEV features from voxelized point clouds to find object centers and regress to 3D bounding boxes.
> To adapt the CluB framework for this comparison, we excluded the query-level enhancement, since CenterPoint does not employ a transformer architecture. Remarkably, our method still achieves higher accuracy (from 67.4% to 68.2%) compared to CenterPoint, **even when utilizing only feature-level enhancement**. We have added these results to the appendix of the revised version to show the universality of the method.
>
>
> | Method      | Vehicle | Pedestrian | Cyclist | mAPH/L2 |
> |-------------|---------|------------|---------|---------|
> | CenterPoint | 67.9    | 65.6       | 68.6    | 67.4    |
> | +CluB       | 68.4    | 66.5       | 69.6    | 68.2    |
>
>
>
>
> [1] Lue Fan et al., Fully Sparse 3D Object Detection, NeurIPS 2022.
>
> [2] Tianwei Yin et al., Center-Based 3D Object Detection and Tracking, CVPR 2021.

---

> > ### Comment · Reviewer_KK49 · 2023-08-20
> > **Raise my score**
> >
> > The authors address most of my concerns and I thus decide to upgrade my score to Borderline Accept

---

### Official Review · Reviewer_BdkH · 2023-07-05

**Soundness:** 3 good
**Presentation:** 3 good
**Contribution:** 3 good
**Rating:** 6
**Confidence:** 4

**Summary:**

The authors explore and analyze the existing LiDAR-based 3D object detection framework, and propose to adopt BEV-based and cluster-based methods to aggregate those features from LiDAR input. The experimental results on Waymo and nuScenes are better compared to the existing methods.

**Strengths:**

1. The task of 3D object detection is very important in the 3D community. Interestingly, the authors propose to extract and combine BEV and cluster-based features.

2. The paper is easy to follow.

3. The authors conduct the experiments on two widely-used datasets, including nuScenes and Waymo.

**Weaknesses:**

1. Computation/memory footprint comparison. The authors didn't make a comparison of their work in terms of memory/speed with the existing 3D detection methods. The time consumption might be large since the proposed method includes transforming the vote cluster to BEV matching and aggregation.

2. Most 3D object detection SOTA is multi-frame based. It would be interesting to see how the multiple frames fitted into the proposed architecture.

**Questions:**

Please refer to the questions that I describe in the Weakness part. I would also consider the rebuttal and other reviews.

**Limitations:**

Yes. The authors presented one of the limitations in terms of computational tradeoffs.

---

> ### Author Rebuttal · Authors · 2023-08-10
>
> Thank you for the thoughtful review. We respond to your comments below
>
> > *Computation/memory footprint comparison.*
>
> Thank you for the suggestion. We have computed the statistics for computation and memory usage of our CluB model and the widely used BEV-based baseline detector Transfusion-L [1] on a single NVIDIA GEFORCE RTX 3090 GPU.
>
> As shown in the table, the computational overhead of CluB is 1.2 times that of the BEV-only baseline in terms of latency. Meanwhile, the FLOPs and memory cost do not go up significantly as the cluster branch primarily involves fully sparse operations on the given point set [2].
>
> |                 | Baseline   | CluB       |
> |-----------------|------------|------------|
> | FLOPs (G)       | 95.71      | 112.35     |
> | Latency (ms)    | 144        | 167        |
> | GPU Memory (MB) | $1.11 × 10^4$ | $1.38 × 10^4$ |
>
>
>
> > *It would be interesting to see how the multiple frames fitted into the proposed architecture.*
>
> Thanks for your constructive feedback. It points out an insightful and crucial direction for our future research. We think it is possible to fit multiple frames into the CluB framework.  We kindly refer you to the PDF (**Figure 2**) for the overview of the framework. Next, we give a detailed illustration of multi-frame-based CluB architecture.
>
> Since our CluB is a query-based 3D detector, we could conveniently model the temporal interaction by utilizing object queries. This is inspired by the object-query-centric temporal modeling architecture of StreamPETR [3]. As illustrated in Figure 2 of the PDF, we first build a memory queue to store the historical object queries. The current queries from our CluB framework are fed into the propagation transformer to interact with historical queries and current BEV features, obtaining temporal and spatial information. The output queries are further used to generate detection results and the top-K queries are pushed into the memory queue. Through the recurrent update of the memory queue, the long-term temporal information is propagated frame by frame. Note that the memory queue follows the first-in, first-out (FIFO) rule.
>
> We would take this constructive suggestion and extend our CluB framework to a multi-frame version in the future.
>
> [1] Xuyang bai et al., TransFusion: Robust LiDAR-Camera Fusion for 3D Object Detection with Transformers, CVPR 2022.
>
> [2] Lue Fan et al., Fully Sparse 3D Object Detection, NeurIPS 2022.
>
> [3] Shihao Wang et al., Exploring Object-Centric Temporal Modeling for Efficient Multi-View 3D Object Detection, arXiv 2023.

---

> > ### Comment · Reviewer_BdkH · 2023-08-21
> > **Authors have addressed most of my concerns.**
> >
> > Thanks for the answers and clarification in the rebuttal, which covered most of my concerns.

---

### Official Review · Reviewer_oemX · 2023-07-06

**Soundness:** 3 good
**Presentation:** 3 good
**Contribution:** 2 fair
**Rating:** 5
**Confidence:** 5

**Summary:**

The paper introduces a new 3D object detection framework called CluB that combines the strengths of BEV-based and cluster-based detectors. CluB effectively integrates the context-aware BEV representation and object-centric cluster representation at both the feature level and the query level.
The proposed method outperforms state-of-the-art methods by a remarkable margin on two prevalent datasets, i.e., Waymo Open Dataset and nuScenes, demonstrating the effectiveness and generality of the proposed method.

**Strengths:**

1. The paper introduces a novel framework that combines the strengths of BEV-based and cluster-based detectors at both the feature level and the query level. The proposed combination method is of a certain degree of novelty.
2. The paper is well-written and easy to follow. The authors provide clear explanations of the proposed method and the experimental results.
3. The paper presents a well-designed and comprehensive experimental evaluation of the proposed CluB framework. The authors provide detailed analysis and comparison with state-of-the-art methods on prevalent datasets.

**Weaknesses:**

1. Limited novelty: While the CluB framework is unique and original in its approach to integrating BEV-based and cluster-based detectors, the paper does not introduce any fundamentally new concepts or techniques. The proposed method is built on existing methods and combines them in a novel way. The novelty is limited for NeurIPS.

2. Limited analysis of real-time performance: The paper does not provide a detailed analysis of the real-time performance of the proposed CluB framework. To address this weakness, the authors could provide a more detailed analysis of the real-time performance of the proposed method and explore ways to make the method more suitable for real-time scenarios.

3. Limited analysis of the combination result of two level: According to the ablation study in table 4. The comprehensive improvement effect of the two levels (3.4%) is greater than the sum of their individual improvement effects (2% + 1.1%). I think this may be the essential reason why the proposed framework is effective, but this issue lacks in-depth analysis.

**Questions:**

1. According to the expression in the paper, Diffused Vote BEV features and Dense BEV features represent the center point semantic information and edge information respectively. But in the design of the Club, Imitation Loss is used to make the two similar. This seems to be contradictory to the previous statement. Can you give some further explanation?
2. As stated in Weakness #3, can you provide a more detailed analysis of the the combination result of two level?
2. Can you provide a more detailed analysis of the computational complexity?
3. Can you provide a more detailed analysis of the impact of the proposed CluB framework on real-time performance?

**Limitations:**

Authors have adequately addressed the limitations.

---

> ### Author Rebuttal · Authors · 2023-08-10
>
> We sincerely appreciate your constructive and detailed comments. We respond to your comments below.
>
> > *According to the expression in the paper, Diffused Vote BEV features and Dense BEV features represent the center point semantic information and edge information respectively. But in the design of the Club, Imitation Loss is used to make the two similar. This seems to be contradictory to the previous statement. Can you give some further explanation?*
> >
> Thanks for your valuable comment. The imitation loss is leveraged to align the center feature between the cluster branch and the BEV branch in the `align-and-fusion` manner, instead of erasing the different information of these two branches. Particularly, as presented in Equation (2) and Equation (3), the valid pixels with imitation loss are within the center region, while no constraint has been added for the edge. In the revision, we will include this explanation in Section 3.3.
>
>
> > *As stated in Weakness #3, can you provide a more detailed analysis of the combination result of two level?*
> >
> Thank you for the thought-provoking comment. It is possible that the overall performance improvement is greater than the summation of their individual improvement. When the two-level enhancements are applied together, the enriched queries are in fact utilized by the enhanced feature from the  feature-level. We provide a more detailed analysis as follows.
>
> Accordingly, we provide results for each category in the following table (Table 1 in the PDF). Compared with BEV-based baseline method (a), the performance improvements are similar over three classes when applying feature-level enhancement. As the features are enhanced in the first step, the improvements for adding queries furthermore are more pronounced, which indicates these two-level enhancements could work synergistically. We observe the same phenomenon you mentioned is mainly on small objects, e.g., 4.0% > 2.3% + 1.4% for pedestrians, which shows that the CluB framework is more effective in capturing fine-grained targets.
> We will add the experimental results and analysis in the future revision.
>
> | Method | Feature-level | Query-level | Vehicle  | Pedestrian | Cyclist  | mAPH/L2  |
> |--------|---------------|-------------|----------|------------|----------|----------|
> | (a)    |               |             | 61.0     | 60.4       | 60.1     | 60.5     |
> | (b)    | √             |             | 62.6 ↑1.6 | 62.7 ↑2.3   | 62.0 ↑1.9 | 62.5 ↑2.0 |
> | (c)    |               | √           | 61.7 ↑0.7 | 61.8 ↑1.4   | 61.4 ↑1.3 | 61.6 ↑1.1 |
> | (d)    | √             | √           | 62.9 ↑1.9 | 64.4 ↑4.0   | 64.5 ↑4.4 | 63.9 ↑3.4 |
>
>
> > *Can you provide a more detailed analysis of the computational complexity?*
> >
> Thank you for the suggestion. The computational overhead of CluB is 1.3 times that of the BEV-only baseline in terms of FLOPs. The computation cost is affordable since the auxiliary cluster branch is built with fully sparse operation, such as sparse instance recognition (SIR) module [1]. Please refer to the table below:
>
> |           | Baseline | CluB   |
> |-----------|----------|--------|
> | FLOPs (G) | 95.71    | 112.35 |
>
>
> >*Can you provide a more detailed analysis of the impact of the proposed CluB framework on real-time performance?*
>
> Thank you for the suggestion. We evaluate the runtime on one NVIDIA GEFORCE RTX 3090. The computational overhead of CluB is 1.2 times that of the BEV-only baseline in terms of latency.  Please see the table below:
>
> |              | Baseline | CluB |
> |--------------|----------|------|
> | Latency (ms) | 144      | 167  |
>
> [1] Lue Fan et al., Fully Sparse 3D Object Detection, NeurIPS 2022.

---

### Official Review · Reviewer_Z2j1 · 2023-07-10

**Soundness:** 2 fair
**Presentation:** 3 good
**Contribution:** 2 fair
**Rating:** 5
**Confidence:** 4

**Summary:**

This paper proposes a CluB framework to improve the accuracy of 3D object detection by taking advantage of both BEV-based and cluster-based paradigms. The motivation is that the cluster features in voting-based cluster method can largely preserve the 3D structure details of each object, thus supplements the weakened center point features in BEV-based convolutional methods. To achieve the above goals, the authors propose a Cluster Feature Diffusion (CFD) module that adaptively diffuses the valid votes on a vote BEV and fuse it with dense BEV features. An imitation loss is also introduced to transfer object-centric knowledge to the BEV branch and encourage the stability of overall representation learning. Meanwhile, a Cluster Query Generation (CQG) module is proposed to enrich the diversity of object queries by using the voting centers from the cluster branch. Extensive experiments are conducted on the Waymo and NuScenes dataset. Both the ablation studies and the comparison with the SOTA methods demonstrate the effectiveness of the proposed method.

**Strengths:**

Overall the paper is well-written and well-structured.
The experiment results are convincing.
The motivation of combining the BEV-based and voting-based method is somewhat novel.
From the technical aspect, although the design of CFD and CQG modules is vulgaris, however it is also effective.

**Weaknesses:**

The demonstration of the class-aware BEV diffusion is not clear. How to leverage the classification results to control the expansion magnitude? I think there should be a formal formula. Figure 3 is also confusing, how to get Class-aware Vote BEV?
There is a lack of discussion of Model Efficiency. It seems that the cluster branch is time-consuming.

**Questions:**

Intuitively, it seems like that the convolution operation performed on feature maps is also like an implicit clustering method that aggregates the surrounding features. Is it necessary to make such big effort to introduce a new cluster branch? If you want to alleviate the phenomenon that a stack of convolution layers will weaken the capability of presenting an object with the center point and reduce structure information, how about using deformable convolution?
Other question, please refer the weakness part above and give your demonstration.

**Limitations:**

None.

---

> ### Author Rebuttal · Authors · 2023-08-10
>
> Thank you for the thoughtful review. We respond to your comments below.
>
> > *The demonstration of the class-aware BEV diffusion is not clear. How to leverage the classification results to control the expansion magnitude? Figure 3 is also confusing, how to get Class-aware Vote BEV?*
>
> Thanks for your valuable comments. The expansion magnitute is adapted according to the predictions by setting different max pooling kernel sizes. Specifically, on Waymo Open Dataset, we set the kernel size to 1, 3 and 5 if a cluster is with maximum likelihood to be pedestrian, cyclist, and vehicle, respectively. Here we adopt the different expansion magnitute since they are in different scales. These details are included in Section 4.1 of the manuscript. We will further emphasize them.
> Besides, we have revised Figure 3 (shown in **Figure 1 of the PDF**) to make it clearer, and elaborate the process of getting Class-aware Vote BEV as follows:
>
> 1. We convert the cluster features to the BEV space and generate vanilla vote BEV features F.
> 2. Since the feature maps are class-agnostic, we merge the classification results (assuming c categories) from the cluster branch into masks { $W_1,W_2, …,W_{c}$ }. The vote BEV feature $F_i$ for each class is then computed using the following equation:
> $F_i=W_i \cdot F$.
> 3. As objects in different classes vary in size, we choose **different kernel sizes according to the classification results of each cluster** (i.e., 1 for pedestrian, 3 for cyclist and 5 for vehicle).
> 4. Based on the defined kernel sizes, we perform max pooling on each per-class vote BEV feature simultaneously, which generates the diffused vote BEV feature of each class $D_i$.
> $D_i=\text{maxpooling}(F_i,\text{kernel size})$
> 5. Finally, **the generated class-aware vote BEV** { $D_1,D_2, …,D_c$ } are together fed into a fully connected layer, which generates the required class-aware vote BEV.
>
> In the final version, we will accordingly revise Section 3.3 to provide a more detailed and intuitive explanation.
>
>
> > *lack of discussion of Model Efficiency.*
>
> Thank you for the suggestion. We present the comparison of model efficiency between our CluB and the BEV-only baseline in terms of latency and FLOPs accordingly. The results are obtained on Waymo Open Dataset with one NVIDIA 3090 GPU.
> The computational overhead of CluB is 1.2 / 1.3  times that of the BEV-only baseline in terms of latency / FLOPs. The cost is affordable since the auxiliary cluster branch is built with fully sparse operations[1]. Detailed comparison is shown in the following table. We will add the experimental results and analysis in the future revision. Thanks again for your advice.
>
> |             | Baseline | CluB   |
> |--------------|-------------------|--------|
> | Latency (ms) | 144               | 167    |
> | FLOPs (G)    | 95.71             | 112.35 |
>
> > *It seems like that the convolution operation performed on feature maps is also like an implicit clustering method that aggregates the surrounding features. Is it necessary to make such big effort to introduce a new cluster branch?*
>
> We agree that the convolution operation can aggregate the surrounding features. However, the filters of convolution have fixed kernel sizes, which means the receptive field might not adapt optimally to objects with different scales and aspect ratios. By contrast, our cluster-based branch groups points (voxels) into clusters with a flexible receptive field. Besides, the center voting operation is leveraged in the cluster branch, which further benefits introducing the object-centric feature.
>
> > *If you want to alleviate the phenomenon that a stack of convolution layers will weaken the capability of presenting an object with the center point and reduce structure information, how about using deformable convolution?*
>
> Thanks for your valuable comments. We agree that it is feasible to exploit deformable convolution for extracting center features, but the performance by leveraging it is unknown to the point cloud domain. To verify it, we conducted an additional experiment. Specifically, we replace all the 2D convolution layers in the BEV backbone with deformable convolution layers. Compared with the baseline (60.5%), this approach leads to inferior detection performance (20.6%). Accordingly, although deformable convolution has been proven to be effective for learning the deformable shape and scale, it barely works when the object center is empty. We will add the experiment and analysis in the revision.
>
> [1] Lue Fan et al., Fully Sparse 3D Object Detection, NeurIPS 2022.

---

### Author Rebuttal · Authors · 2023-08-10

We sincerely thank the reviewers for their detailed and insightful comments, as well as the favorable recommendations. We also thank the area chair for your time and efforts in handling our paper. We appreciate the positive comments, e.g., "well-written" and "novel motivation" from Reviewer Z2j1, "well-designed and comprehensive evaluation" from Reviewer oemX, "easy to follow"  from Reviewer BdkH, "interesting hypothesis" and "compact structure" from Reviewer KK49, "well-motivated","concise illustration" and "well-ablated" from Reviewer UKd9.

Following the valuable comments and suggestions, we carefully revised the manuscript and updated the suggested experiments on the supplementary material. We hereby refer the reviewers to the detailed responses to each comment. We also upload a PDF with tables and figures due to the limited characters of each response. We provide a summary of changes presented in the attached PDF.

- Tables
  - Table 1: As suggested by Reviewer oemX,  we provide a more detailed comparison on the effect of the two-level enhancement in CluB on individual classes, showing that the two-level enhancements could work synergistically.
  - Table 2: As suggested by Reviewer KK49, we conduct an ablation study on different ways to aggregate the features to the center of the object, showing the effectiveness of cluster representation with its adaptable receptive.
  - Table 3: As suggested by Reviewer KK49, we show the experimental result of applying CluB framework on the different BEV-based detector,  which demonstrates the universality of our method.
  - Table 4: As suggested by Reviewer UKd9, we conduct an ablation study on the effect of the two kinds of object queries for the transformer decoder, showing the two have respective advantages in query initialization for the decoder.
  - Table 5: As suggested by Reviewer UKd9, we present part of the revised Table 1 of the manuscript.

- Figures
  - Figure 1: As suggested by Reviewer Z2j1, we provide a detailed illustration of the Cluster Feature Diffusion (CFD) module, which is carefully revised based on the Figure 3 of the manuscript.
  - Figure 2: As suggested by Reviewer BdkH, we give an illustration on the possible multi-frame version of our proposed CluB, which is inspired by the recent work named StreamPETR.

We look forward to discussing with you over the next few days.

---

### Decision · Program_Chairs · 2023-09-21

**Decision:**

Accept (poster)

**Comment:**

The paper studies how to combine the strengths of BEV-based and cluster-based representations for 3D object detection. Reviewers generally found the problem and insight interesting and novel, the proposed approach effective, and the experiments comprehensive and solid. Meanwhile, reviewers raised several technical questions.

After the rebuttal, many of the reviewers' questions are addressed. Several reviewers increased the ratings. The paper now has a positive rating: average 5.4 (5, 5, 5, 6, 6). The AC thus recommends an acceptance. The AC suggests that the authors incorporate the detailed rebuttal into the final version.